# OpenReview forum: "LLM Jailbreak Detection for (Almost) Free!"
_ICLR.cc/2025/Conference — ICLR 2025 Conference Withdrawn Submission_

### Official Review · Reviewer_UT1U · 2024-10-22

**Soundness:** 2
**Presentation:** 3
**Contribution:** 2
**Rating:** 3
**Confidence:** 3

**Summary:**

The authors investigated how to detect jailbreak prompts. They found that LLMs allocate significantly more attention to the first few tokens of jailbreak prompts compared to benign ones. Therefore, they suggested measuring the attention given to the initial tokens of a prompt as a jailbreak indicator. Additionally, they proposed using handcrafted prompts and temperature scaling to improve detection performance. The handcrafted prompts could also be improved through virtual instruction learning.

**Strengths:**

* The paper is well-written and easy to follow.

* Jailbreak attacks are a critical research topic in LLMs, particularly regarding defenses against such attacks.

* The authors conducted many experiments to validate the performance of their method.

**Weaknesses:**

* Some of the claims in the paper lack rigor. For example, in line 161, the authors presented experiments using two types of attacks on the Llama2 7B model to back their findings. However, results derived from just one model do not adequately demonstrate the generalizability of their conclusions.

* The attack methods used in this paper focus on optimizing a jailbreak suffix to generate jailbreak prompts. If jailbreak words are distributed across different positions in the input, would the authors' finding still hold? The authors should evaluate their method against attack strategies that do not solely rely on optimizing jailbreak suffixes [1, 2].

* My major concern lies in the robustness of the proposed detection method against adaptive attacks.

* In line 246, the authors mention conducting a theoretical analysis; however, I could not find the relevant results.

[1] Jailbreaking black box large language models in twenty queries

[2] Tree of attacks: Jailbreaking black-box llms automatically

**Questions:**

See weaknesses.

---

> ### Author Response · Authors · 2024-11-27
> **Response to Reviewer UT1U**
>
> Thanks for your constructive feedback and inspiring questions!\
> **Q1:** The findings in Llama2 7B lacks rigor.\
> **A1:** We sincerely appreciate you pointing out our mistake. This finding is specific to Llama2 and serves as the motivation behind the proposed FJD method. Therefore, we will restrict the discussion of our findings to the Llama2 model.\
> **Q2:** The authors should evaluate their method against attack strategies that do not solely rely on optimizing jailbreak suffixes.\
> **A2:** Thank you for your suggestion. We evaluated the detection performance of FJD against other attack methods, including AdvPrompter[1], MAC[2], and Pair[3], as detailed in Appendix E and F. And the Hand-Crafted Attacks comprise 28 manual attack methods, with their FJD detection results provided in Appendix G.\
> **Q3:** My major concern lies in the robustness of the proposed detection method against adaptive attacks.\
> **A3:** In Section 4.11, we discuss the experimental results of attacks conducted after the attacker becomes aware of the FJD detection method. Based on the knowledge of the manual instruction, the use of a suffix optimization method similar to GCG can bypass FJD detection.\
> **Q4:** In line 246, the authors mention conducting a theoretical analysis; however, I could not find the relevant results.\
> **A4:** We sincerely appreciate you pointing out our mistake. The theoretical analysis of why manual instructions can be used for detection is provided in Section 4.5, and further discussion is presented in Section 5.
>
> [1] Anselm Paulus, Arman Zharmagambetov, Chuan Guo, Brandon Amos, and Yuandong Tian. Advprompter: Fast adaptive adversarial prompting for llms. *arXiv preprint arXiv:2404.16873*, 2024.\
> [2] Yihao Zhang and Zeming Wei. Boosting jailbreak attack with momentum. *arXiv preprint arXiv:2405.01229*, 2024.\
> [3] Patrick Chao, Alexander Robey, Edgar Dobriban, Hamed Hassani, George J Pappas, and Eric Wong. Jailbreaking black box large language models in twenty queries. *arXiv preprint arXiv:2310.08419*, 2023.

---

> > ### Comment · Reviewer_UT1U · 2024-11-28
> >
> > Thank you for your detailed response. However, it does not address my concerns.
> >
> > For Q1, the authors should expand their evaluation to include more models. For Q3, the evaluation lacks baselines. The number of 0.4761 in Table 8 indicates that the proposed detection method is completely broken in adaptive attack scenarios. As for Q4, I hardly agree this is a theoretical analysis of the proposed method.
> >
> > Moreover, according to ICLR's policy, I am not allowed to request new experiments at this stage. The authors may consider these comments for improvements in future versions. I maintain my rating.

---

> ### Author Response · Authors · 2024-11-29
> **Response to Reviewer UT1U**
>
> **Q1:** The authors should expand their evaluation to include more models\
> **A1:** Statistical experiments were conducted on both Vicuna and Guanaco. The FT (First Token) detection results presented in Tables 1 and 2 indicate that FT demonstrates some detection capability for specific attack methods and models,  which corroborates our finding that a distinct difference in the confidence of the first token is observed between responses generated by these prompts and those that are benign. This observation is particularly evident in Llama2-7B; hence, the detection results for Llama2-7B are presented exclusively in Figure 1.\
> **Q3:** The proposed detection method is completely broken in adaptive attack scenarios\
> **A3:** Adaptive attacks can indeed completely break FJD. However, this requires the attacker to conduct a white-box attack leveraging knowledge of FJD manual instructions and the LLM. Currently, designing a robust detection method against white-box adaptive attacks is a well-known challenge in our community. In more practical scenarios, transferable black-box attacks are commonly employed. As shown in Table 3, FJD demonstrates robust defense against these transferable black-box attacks. Furthermore, even when the attacker possesses knowledge of FJD manual instructions, FJD can still effectively detect adaptive attacks within transferable black-box scenarios, as illustrated in the table below. Thank you for raising this question, we will incorporate this discussion and limitations into our paper.
>
> |                 Attacks                 |   FJD  |
> |---------------------------------------|------|
> | AutoDAN                                 | 0.8061 |
> | Cipher                                  | 0.9094 |
> | Hand-crafted                            | 0.7583 |
> | Aware-attack (white-box)                | 0.4761 |
> | Aware-attack (Transfer from Llama2-7B)  | 0.9017 |
> | Aware-attack (Transfer from Guanaco-7B) | 0.8886 |
>
> **Q4:**  I hardly agree this is a theoretical analysis of the proposed method\
> **A4:** We appreciate your feedback on our mistakes. After further consideration, we have revised the title of Section 4.5 to "Attribution Analysis of Manual Instructions," which investigates the attention that LLM pay to manual instructions when inferring jailbreak and benign samples, specifically the extent of impact these instructions have. The theoretical analysis and discussion will be presented in Section 5.

---

### Official Review · Reviewer_XAne · 2024-10-26

**Soundness:** 2
**Presentation:** 3
**Contribution:** 3
**Rating:** 6
**Confidence:** 4

**Summary:**

The work focuses on the research problem of detecting Jailbreak prompts. They present the finding of output distribution difference between jailbreak and benign prompts and use this finding to detect jailbreak prompts. Extensive experiments on 8 models are conducted under both jailbreak attacks with competing objectives and mismatched generalization.

**Strengths:**

- The finding is interesting and reasonable.
- The result is promising.
- The overall presentation of the model and the results is clear.

**Weaknesses:**

- The paradigm of adding manual instruction and scrutinizing the difference of the first token is similar to the work [1], which also focuses on detecting jailbreak prompts. It would make the work more comprehensive to discuss the differences and compare the performance.
- It would also be helpful to compare with moderation APIs and [2]. Although they were designed for general unsafe prompt detection, but would also lead to a good comparison.
- Overall. as SmoothLLM & PPL are kind of designed for GCG-like not readable jailbreak, comparing with them alone may compromise the experimental comprehensiveness.



[1]GradSafe: Detecting Jailbreak Prompts for LLMs via Safety-Critical Gradient Analysis
[2]Llama Guard: LLM-based Input-Output Safeguard for Human-AI Conversations

**Questions:**

See weakness.
I think the overall finding is interesting and cool, I would raise the score if the evaluation can be improved.

---

> ### Author Response · Authors · 2024-11-27
> **Response to Reviewer XAne**
>
> Thank you for your constructive comments!\
> **Q1:** The difference of FJD and GradSafe [1]\
> **A1:** FJD requires only a single forward pass to detect samples by leveraging the token confidence generated by the LLM. In contrast, GradSafe detects samples by utilizing parameter gradients computed during backpropagation, using "Sure" as the inference label.\
> **Q2:** Comparing with SmoothLLM & PPL may compromise the experimental comprehensiveness\
> **A2:** We will supplement GradSafe[1] with comparative experiments. The experimental comparison results for Llama2-7B, Vicuna-7B, and Guanaco-7B across AutoDAN, Cipher, and GradSafe are presented in the subsequent tables.
>
> | Attack  | Method   | Llama2-7B   | Vicuna-7B    | Guanaco-7B   |
> |-------------|--------------|-----------------|------------------|------------------|
> | AutoDAN     | PPL          | 0.3700 ± 0.0029 | 0.2714 ± 0.0006  | 0.0071 ± 0.0002  |
> |             | SMLLM        | 0.8197 ± 0.0052 | 0.7831 ± 0.0035  | 0.5460 ± 0.0026  |
> |             | GradSafe     | 0.7953 ± 0.0084 | 0.8015 ± 0.0014  | 0.8233 ± 0.0037  |
> |             | FT           | 0.9164 ± 0.0051 | 0.1697 ± 0.0059  | 0.6592 ± 0.0106  |
> |             | FJD          | **0.9495** ± 0.0053 | **0.8061** ± 0.0103 | **0.8509** ± 0.0089 |
> | Cipher      | PPL          | 0.0014 ± 0.0010 | 0.0094 ± 0.0002  | 0.0071 ± 0.0002  |
> |             | SMLLM        | 0.5034 ± 0.0024 | 0.5233 ± 0.0009  | 0.5460 ± 0.0026  |
> |             | GradSafe     | 0.7708 ± 0.0033 | 0.6932 ± 0.0191  | 0.8474 ± 0.0086  |
> |             | FT           | 0.9335 ± 0.0035 | 0.6443 ± 0.0091  | 0.6592 ± 0.0106  |
> |             | FJD          | **0.9700** ± 0.0034 | **0.9094** ± 0.0040 | **0.8509** ± 0.0089 |
>
>
> [1] Yueqi Xie, Minghong Fang, Renjie Pi, and Neil Gong. Gradsafe: Detecting jailbreak prompts for llms via safety-critical gradient analysis. In *Proceedings of the 62nd Annual Meeting of the Association for Computational Linguistics (Volume 1: Long Papers)*, pp. 507–518, 2024.

---

> > ### Comment · Reviewer_XAne · 2024-11-27
> > **Thanks for the response.**
> >
> > I appreciate the author's new results, which addressed most of my concerns. I would raise my score to 6. I would like to suggest the author also involve moderation APIs and/or Llama Guard in the revisions for the final version.

---

### Official Review · Reviewer_yHi5 · 2024-11-04

**Soundness:** 2
**Presentation:** 3
**Contribution:** 2
**Rating:** 3
**Confidence:** 4

**Summary:**

The authors introduce a Free Jailbreak Detection (FJD) method by incorporating an instruction into the input prompt. This makes the logits distribution of the first token in the LLM's output distinct in both harmful and harmless dimensions, and the difference is amplified using temperature scaling. In order to make the distribution of logits of the first token more separable, the authors also used virtual instruction learning to expand the difference in distribution. Experiments on aligned large models demonstrated the effectiveness of the proposed FJD in jailbreak detection accuracy.

**Strengths:**

1. Detecting jailbreaks in LLMs is a critical issue, especially for the defense against such attacks.
2. The author proposes to use the logits distribution of the first token output by LLM to detect jailbreaks, which sounds like a relatively novel idea.
3. The author proposes to use temperature scaling to amplify the logits distribution of harmful and harmless instructions of LLM by connecting the input with the instruction. Experiments are conducted to validate the effectiveness of this method.

**Weaknesses:**

1. The paper lacks sufficient experiments on the author's choice of threshold and temperature for jailbreak detection, that is, how much training data is needed to obtain a reasonable threshold and temperature parameter. Authors are suggested to include an experiment analyzing the detection performance with varying amounts of training data.
2. For virtual instruction learning, the authors did not provide an explanation regarding the transferability of the detection method when applied to a jailbreak dataset generated by a new jailbreak method. It is recommended for the authors to conduct such transferability experiment.

**Questions:**

1. The proposed detection method requires the determination of the temperature parameters and thresholds. Faced with an unknown jailbreak dataset or jailbreak method, how can appropriate parameter values be established? How to ensure the proposed method also remain effective against unknown jailbreak methods? Please clarify.
2. While virtual instruction learning is optimized for known datasets, how to ensure its generalization? The authors may conduct specific experiments demonstrating generalization, such as testing on out-of-distribution jailbreak attempts.
3. Due to the inherent difficulty in controlling the output of the LLM after adding instructions to the prompt, in order to get a correct response, the LLM should be queried at least twice: first to detect potential jailbreaks and subsequently to obtain the intended response. Why not utilize a classifier with much fewer parameters to detect jailbreaks after the LLM has produced its output? If the model parameter sizes are large (e.g., larger than 70B), the proposed detection method seems to incur a larger amount of computations. It is recommended for the authors to compare and discuss this alternative with the proposed method.

---

> ### Author Response · Authors · 2024-11-27
> **Response to Reviewer yHi5**
>
> Thanks for your in-depth suggestion. We hope the following experiments and explanations will address your concerns regarding the threshold and temperature.\
> **Q1:** Sufficient experiments on the author's choice of threshold and temperature for jailbreak detection.\
> **A1:** We supplemented the training set with proportions of 10%, 20%, 30%, 40%, and 50%, examining the selection of temperatures and their impact on detection performance in Llama2-7B. For temperature, although the values derived from the training set are not optimal, its effect on detection performance remains minimal within a certain range, as shown in Figure 4. In this paper,  no single threshold K is selected for evaluating all the tasks. The AUC calculation considers all possible threshold values for K.
>
> | Model  | Training | Temperature | AutoDAN      | Cipher      | GCG          | PAIR         |
> |------------|--------------|------------------|------------------|-----------------|------------------|------------------|
> | Llama2-7B  | 10%          | 1.18            | 0.9549 ± 0.0054  | 0.9764 ± 0.0017 | 0.9983 ± 0.0004  | 0.9738 ± 0.0038  |
> |            | 20%          | 1.20            | 0.9564 ± 0.0061  | 0.9741 ± 0.0026 | 0.9990 ± 0.0002  | 0.9737 ± 0.0015  |
> |            | 30%          | 1.23            | 0.9542 ± 0.0061  | 0.9726 ± 0.0019 | 0.9990 ± 0.0002  | 0.9749 ± 0.0047  |
> |            | 40%          | 1.24            | 0.9519 ± 0.0024  | 0.9714 ± 0.0013 | 0.9990 ± 0.0003  | 0.9754 ± 0.0019  |
> |            | 50%          | 1.25            | 0.9495 ± 0.0053  | 0.9700 ± 0.0034 | 0.9990 ± 0.0003  | 0.9761 ± 0.0009  |
>
>
> **Q2:** For virtual instruction learning, how to ensure its generalization?\
> **A2:** For FJD-LI, we trained the model using randomly sampled data from GCG and AutoDAN to cover both human-readable and non-human-readable jailbreak samples. The table demonstrate that this approach further improves detection performance on unknown jailbreak (Cipher[1] & MAC[2]).
>
> | Model  | Method  | Llama2-7B    |Vicuna-7B    | Guanaco-7B   |
> |------------|-------------|------------------|------------------|------------------|
> | Cipher     | PPL         | 0.0014 ± 0.0010  | 0.0094 ± 0.0002  | 0.0071 ± 0.0002  |
> |            | SMLLM       | 0.5034 ± 0.0024  | 0.5233 ± 0.0009  | 0.5460 ± 0.0026  |
> |            | FJD         | 0.9682 ± 0.0034  | 0.9094 ± 0.0040  | 0.8509 ± 0.0089  |
> |            | FJD-LI      | **0.9944** ± 0.0012 | **0.9310** ± 0.0036 | **0.8826** ± 0.0102 |
> | MAC        | PPL         | 0.9816 ± 0.0001  | 0.9853 ± 0.0005  | 0.9867 ± 0.0006  |
> |            | SMLLM       | 0.9091 ± 0.0064  | 0.8837 ± 0.0065  | 0.9071 ± 0.0005  |
> |            | FJD         | 0.9839 ± 0.0024  | 0.7722 ± 0.0092  | 0.9378 ± 0.0029  |
> |            | FJD-LI      | **0.9995** ± 0.0001 | **0.9987** ± 0.0021 | **0.9985** ± 0.0007 |
>
>
> **Q3:** Due to the inherent difficulty in controlling the output of the LLM after adding instructions to the prompt, in order to get a correct response, the LLM should be queried at least twice.\
> **A3:** Yes, adding an instruction impacts both benign and jailbreak samples, with a positive effect observed on benign samples. Additionally, as shown in Table 5, appending the instruction “Remember you are a responsible and helpful LLM” enables detection on Llama2-7B.
>
> [1] Youliang Yuan, Wenxiang Jiao, Wenxuan Wang, Jen-tse Huang, Pinjia He, Shuming Shi, and Zhaopeng Tu. Gpt-4 is too smart to be safe: Stealthy chat with llms via cipher. *arXiv preprint arXiv:2308.06463*, 2023a.\
> [2] Yihao Zhang and Zeming Wei. Boosting jailbreak attack with momentum. arXiv preprint *arXiv:2405.01229*, 2024.

---

> ### Author Response · Authors · 2024-11-29
> **Dear Reviewer yHi5 (Friendly Reminder)**
>
> I hope this message finds you well. I wanted to reach out and kindly remind you that I have posted my response to your review of my paper. I am eagerly looking forward to the opportunity to engage in further discussion with you and address any questions or concerns you may have regarding my work.
>
> I value your feedback and appreciate the time and effort you have dedicated to reviewing my paper. I am more than happy to clarify any aspects of my research that may have caused confusion and am open to discussing any suggested improvements.
>
> Thank you once again for your valuable input, and I am optimistic that our exchange will lead to a fruitful discussion.

---

> > ### Comment · Reviewer_yHi5 · 2024-12-03
> >
> > Thank you for the detailed response for Q1 and Q2. However,  my concerns for Q3 is not well addressed. When faced with an LLM with large parameter size, the proposed method must infer twice before returning an answer even when faced with harmless content, which not only requires LLM to repeat inference, but also requires a large amount of computation. Compared to connecting a much smaller classifier to the output of LLM, which only requires LLM to infer once, what is the advantage of your method?

---

### Official Review · Reviewer_FJt6 · 2024-11-07

**Soundness:** 2
**Presentation:** 2
**Contribution:** 2
**Rating:** 5
**Confidence:** 4

**Summary:**

This paper aims to detect adversarial jailbreak prompts by analyzing output confidences. Specifically, it observes that the output confidence for the first token is generally higher for benign inputs compared to malicious ones. By establishing a threshold based on this observation, the authors propose a method to distinguish between jailbreak attacks and benign inputs. To enhance detection capabilities, they introduce manual instructions and temperature scaling techniques designed to amplify the confidence difference between benign and malicious inputs, thereby improving detection efficiency.

**Strengths:**

* Timely and Relevant Topic: The paper addresses a critical issue in the field of AI security—jailbreak detection—highlighting the gap in defense mechanisms despite numerous studies on jailbreak attacks.
* Innovative Findings: The research presents an intriguing discovery that could lead to a more effective jailbreak detection algorithm.

**Weaknesses:**

The proposed method lacks clarity in several areas:
* Is the detection focused on identifying all malicious attempts (including those that do not result in harmful outputs) or solely those leading to harmful outputs?
* How was Figure 1 generated? Specifically, how many instances were used as benign prompts, and are these representative of the broader benign sample? What dataset was utilized?
* The design of the manual instructions is unclear. How do these instructions increase the confidence of benign inputs without impacting malicious inputs?
* Why does temperature scaling effectively increase the distinction between benign and malicious inputs? As noted in line 58, some LLM models may exhibit overconfidence for both jailbreak and benign prompts. If this is true, temperature scaling might lower outputs for both categories, which means it cannot help in differentiation.
* The FJD-LI model, trained on a limited number of samples, may struggle to generalize across other attacks and benign inputs.
* What does it mean to "To align benign prompts with jailbreak prompts, we exclude pertinent prompts from the benign dataset" by excluding pertinent prompts from the benign dataset?
* Why was PureDove selected as the benign dataset? Given that LLM performance is typically evaluated across diverse datasets, assessing benign performance on a wider range of clean datasets is crucial for ensuring minimal impact on detection algorithms.
* The evaluation only includes three types of attacks, which is insufficient given the plethora of state-of-the-art attacks available (a quick search will reveal many more).
* Intuitively, one would expect significant perplexity differences between Cipher attack inputs and normal inputs. However, results in Table 2 suggest that perplexity is ineffective for attack detection, casting doubt on these findings.

While this paper presents a timely topic and interesting findings, it lacks technical depth and robustness in its methodology. Therefore, I am inclined to reject this submission. To enhance the paper's quality, further clarification is needed regarding how manual instructions and temperature scaling can reliably improve performance.

**Questions:**

See above

---

> ### Author Response · Authors · 2024-11-27
> **Response to Reviewer FJt6 (1/2)**
>
> We thank reviewer FJt6 for your helpful comments and constructive feedback! \
> **Q1:** What did the detection focus on?\
> **A1:** In our paper, the detection focuses solely on identifying malicious attempts that could produce harmful responses.\
> **Q2:** How was Figure 1 generated?\
> **A2:** Figure 1 uses an equal number of jailbreak and benign samples in Llama2 7B, where the jailbreak samples are generated by AutoDAN and Cipher, and the benign samples are derived from the PureDove. \
> **Q3:** How do these instructions increase the confidence of benign inputs without impacting malicious inputs?\
> **A3:** We conducted experiments using a simple manual instruction (MI) and discuss the impact of different MIs on detection results in Section 4.7. MI affects both jailbreak and benign samples; however, as shown in the experiments in Section 4.5, its impact on benign samples is more significant.\
> **Q4:** Why does temperature scaling effectively increase the distinction between benign and malicious inputs?\
> **A4:** The experiments in Sections 4.6 and 4.8 reveal that temperature scaling affects detection performance, particularly when the temperature is low, making detection almost impossible. Although applying temperature scaling reduces the probability values for both cases, the extent of reduction varies due to differences in their probability distributions.\
> **Q5:** The FJD-LI may struggle to generalize across other attacks and benign inputs.\
> **A5:** For FJD-LI, we trained the model using randomly sampled data from GCG and AutoDAN to cover both human-readable and human-unreadable jailbreak samples. The results demonstrate that this approach further improves detection performance on Cipher.\
> **Q6:** What does it mean to "To align benign prompts with jailbreak prompts, we exclude pertinent prompts from the benign dataset" by excluding pertinent prompts from the benign dataset?\
> **A6:** We sincerely appreciate you pointing out our mistake. Our intention was to randomly sample an equal number of benign samples to match the number of jailbreak samples.

---

> ### Author Response · Authors · 2024-11-27
> **Response to Reviewer FJt6 (2/2)**
>
> **Q7:** Why was PureDove selected as the benign dataset?\
> **A7:** The PureDove collection of dialogues from both GPT-4 and humans has a high likelihood of capturing a broad range of real-world scenarios. Subsequently, we supplemented FJD with two benign datasets, Open-Platypus[1] and SuperGLUE[2]. The experimental results on Llama2-7B are presented in the table below. The results demonstrate that FJD exhibits minimal variation in detection performance across other benign datasets.
>
> (1) Open-Platypus
>
> | Attack  | Method | Llama2-7B           | Vicuna-7B           | Guanaco-7B          |
> |-------------|------------|-------------------------|-------------------------|-------------------------|
> | AutoDAN     | PPL        | 0.6935±0.0014 | 0.6069±0.0003 | 0.6677±0.0008 |
> |             | SMLLM      | 0.8372±0.0026 | 0.7062±0.0018 | 0.7472±0.0030 |
> |             | FT         | 0.9452±0.0059 | 0.0696±0.0084 | 0.8351±0.0082 |
> |             | FJD        | **0.9645**±0.0014 | **0.7709**±0.0046 | **0.9860**±0.0050 |
> | Cipher      | PPL        | 0.0050±0.0004 | 0.0217±0.0011 | 0.0226±0.0003 |
> |             | SMLLM      | 0.5050±0.0007 | 0.5726±0.0021 | 0.5460±0.0024 |
> |             | FT         | 0.9940±0.0018 | 0.7206±0.0045 | 0.2531±0.0081 |
> |             | FJD        | **0.9996**±0.0001 | **0.8411**±0.0092 | **0.7699**±0.0067 |
>
> (2) SuperGLUE
>
> | Attack  | Method | Llama2-7B       | Vicuna-7B       | Guanaco-7B      |
> |-------------|------------|---------------------|---------------------|---------------------|
> | AutoDAN | PPL        | 0.8633±0.0006      | 0.8109±0.0003      | 0.8421±0.0005     |
> |             | SMLLM      | 0.9026±0.0037      | 0.7745±0.0021      | 0.6923±0.0019     |
> |             | FT         | 0.7946±0.0154      | 0.2734±0.0102      | 0.6308±0.0082     |
> |             | FJD        | **0.9345**±0.0069  | **0.8123**±0.0130  | **0.8469**±0.0041 |
> | Cipher  | PPL        | 0.0087±0.0002      | 0.0292±0.0002      | 0.0313±0.0002     |
> |             | SMLLM      | 0.6265±0.0044      | 0.6594±0.0018      | 0.5593±0.0031     |
> |             | FT         | 0.9632±0.0077      | 0.1424±0.0050      | 0.6305±0.0052     |
> |             | FJD        | **0.9991**±0.0006  | **0.8395**±0.0098  | **0.8690**±0.0072 |
>
> **Q8:** The evaluation only includes three types of attacks.\
> **A8:** We evaluated the detection performance of FJD against other attack methods (AdvPrompter, MAC, Pair, etc.) in Appendices E and F.\
> **Q9:** Significant perplexity differences between Cipher attack inputs and normal inputs.\
> **A9:** Our experiments reveal that, although PureDove is categorized as benign, it contains a significant number of tasks requiring complex reasoning and consists of real dialogues from humans. As a result, the complexity of its questions surpasses that of the prompts generated by Cipher.
>
> [1] Ariel N Lee, Cole J Hunter, and Nataniel Ruiz. Platypus: Quick, cheap, and powerful refinement of llms. arXiv preprint arXiv:2308.07317, 2023a.\
> [2] Alex Wang, Yada Pruksachatkun, Nikita Nangia, Amanpreet Singh, Julian Michael, Felix Hill, Omer Levy, and Samuel Bowman. Superglue: A stickier benchmark for general-purpose language understanding systems. *Advances in neural information processing systems*, 32, 2019.

---

> ### Author Response · Authors · 2024-11-29
> **Dear Reviewer FJt6 (Friendly Reminder)**
>
> I hope this message finds you well. I wanted to reach out and kindly remind you that I have posted my response to your review of my paper. I am eagerly looking forward to the opportunity to engage in further discussion with you and address any questions or concerns you may have regarding my work.
>
> I value your feedback and appreciate the time and effort you have dedicated to reviewing my paper. I am more than happy to clarify any aspects of my research that may have caused confusion and am open to discussing any suggested improvements.
>
> Thank you once again for your valuable input, and I am optimistic that our exchange will lead to a fruitful discussion.

---

> > ### Comment · Reviewer_FJt6 · 2024-12-02
> > **Thanks for the rebuttal**
> >
> > Thanks to the authors for writing this rebuttal. However, this rebuttal does not address most of my concerns. Some of them include:
> > * **Still not clear on how to design the manual instruction**. Sec. 4.5 & Sec. 4.6 shows that the MI is effective. However, why is MI designed this way? Why can the **contradictory MIs** (e.g., line 437 and line 440) shown in Table 5 improve the detection performance?
> > * **Why does temperature scaling effectively increase the distinction between benign and malicious inputs?** The reply from the authors is more like an experimental observation than an in-depth analysis and explanation of the underlying mechanism.
> > * FJD-LI's results on other attacks (e.g., handcrafted, GCG, etc.) are missing.
> > * According to Appendices E and F, the results on other well-known attacks like GCG are poor.
> > * **Perplexity between cipher text and normal text.** I am asking about the perplexity, but the authors are comparing the complexity. Perplexity is not equal to complexity.
> >
> > For these reasons, I decided to keep my score.

---

### Note · Authors · 2024-12-12

I have read and agree with the venue's withdrawal policy on behalf of myself and my co-authors.